# Platelets Boost Recruitment of CD133^+^ Bone Marrow Stem Cells to Endothelium and the Rodent Liver—The Role of P-Selectin/PSGL-1 Interactions

**DOI:** 10.3390/ijms21176431

**Published:** 2020-09-03

**Authors:** Nadja Lehwald, Constanze Duhme, Iryna Pinchuk, Julian Kirchner, Kristina Wieferich, Moritz Schmelzle, Kerstin Jurk, Beatrice A. Windmöller, Wolfgang Hübner, Bernhard Homey, Johannes Bode, Ralf Kubitz, Tahar Benhidjeb, Martin Krüger, Simon C. Robson, Wolfram T. Knoefel, Beate E. Kehrel, Jan Schulte am Esch

**Affiliations:** 1Department of Surgery A, University Hospital Duesseldorf, 40225 Duesseldorf, Germany; nadja.lehwald@med.uni-duesseldorf.de (N.L.); conny.duhme@googlemail.com (C.D.); IrynaPinchuk@web.de (I.P.); Kristina.wieferich@web.de (K.W.); knoefel@uni-duesseldorf.de (W.T.K.); 2Department of Diagnostic and Interventional Radiology, University Hospital Duesseldorf, 40225 Duesseldorf, Germany; Julian.Kirchner@med.uni-duesseldorf.de; 3Department of Surgery, Charite, 10117 Berlin, Germany; moritz.schmelzle@charite.de; 4Center for Thrombosis and Hemostasis, Johannes Gutenberg-University, 55122 Mainz, Germany; Kerstin.Jurk@unimedizin-mainz.de; 5Department of Anesthesiology Intensive Care and Pain Medicine, Experimental and Clinical Hemostasis, University of Muenster, 48149 Muenster, Germany; haemostasis.research@uni-muenster.de; 6Department of Cell Biology, Bielefeld University, Universitätsstrasse 25, 33615 Bielefeld, Germany; beatrice.windmoeller@uni-bielefeld.de; 7Biomolecular Photonics, Department of Physics, University of Bielefeld, Universitätsstrasse 25, 33615 Bielefeld, Germany; whuebner@physik.uni-bielefeld.de; 8Department of Dermatology, University Hospital Duesseldorf, 40225 Duesseldorf, Germany; Bernhard.Homey@med.uni-duesseldorf.de; 9Department of Gastroenterology, University Hospital Duesseldorf, 40225 Duesseldorf, Germany; Johannes.Bode@med.uni-duesseldorf.de (J.B.); ralf.kubitz@bethanienmoers.de (R.K.); 10Center of Visceral Medicine, Department of General and Visceral Surgery, Protestant Hospital of Bethel Foundation, 33617 Bielefeld, Germany; tahar.benhidjeb@yahoo.de; 11Center of Visceral Medicine, Department of Gastroenterology and Internal Medicine, Protestant Hospital of Bethel Foundation, 33617 Bielefeld, Germany; martin.krueger@evkb.de; 12The Transplant Institute and Division of Gastroenterology, Beth Israel Deaconess Medical Center, Harvard University, Boston, MA 02215, USA; srobson@bidmc.harvard.edu

**Keywords:** platelets, bone marrow stem cells, CD133, liver regeneration, endothelial cells, P-selectin

## Abstract

We previously demonstrated that clinical administration of mobilized CD133^+^ bone marrow stem cells (BMSC) accelerates hepatic regeneration. Here, we investigated the potential of platelets to modulate CD133^+^BMSC homing to hepatic endothelial cells and sequestration to warm ischemic livers. Modulatory effects of platelets on the adhesion of CD133^+^BMSC to human and mouse liver-sinusoidal- and micro- endothelial cells (EC) respectively were evaluated in in vitro co-culture systems. CD133^+^BMSC adhesion to all types of EC were increased in the presence of platelets under shear stress. This platelet effect was mostly diminished by antagonization of P-selectin and its ligand P-Selectin-Glyco-Ligand-1 (PSGL-1). Inhibition of PECAM-1 as well as SDF-1 receptor CXCR4 had no such effect. In a model of the isolated reperfused rat liver subsequent to warm ischemia, the co-infusion of platelets augmented CD133^+^BMSC homing to the injured liver with heightened transmigration towards the extra sinusoidal space when compared to perfusion conditions without platelets. Extravascular co-localization of CD133^+^BMSC with hepatocytes was confirmed by confocal microscopy. We demonstrated an enhancing effect of platelets on CD133^+^BMSC homing to and transmigrating along hepatic EC putatively depending on PSGL-1 and P-selectin. Our insights suggest a new mechanism of platelets to augment stem cell dependent hepatic repair.

## 1. Introduction

We have previously shown that CD133^+^ bone marrow stem cells (BMSC) can support hepatic repair in a preclinical [1] as well as in a clinical setting of liver resection and regeneration [2,3,4,5]. We previously demonstrated the mobilization of CD133^+^BMSC following extensive liver resection. The latter seems to be both hepatocyte growth factor and stroma-derived factor-1 (SDF-1)-mediated [6]. Others showed an important role of hematopoietic CD133^+^ stem cells interacting with platelets resulting in increased SDF-1 expression [7]. Interestingly, we recently demonstrated CD133^+^BMSC when co-incubated with platelets to control their thrombogenic responses in a CD39-dependent manner [8]. Furthermore, a major role for platelets was described in the mediation of hepatic injury [9,10,11]. There is increasing evidence that points to a crucial complex orchestrated role for platelets in various preclinical and clinical scenarios of hepatic regeneration following injury and resection, respectively [12,13,14,15,16,17,18,19]. Interactions of platelets with liver sinusoidal endothelial cells (LSEC) and release of ADP, serotonin and other platelet growth factors seem to have crucial impacts in regulating the early phase of hepatic regeneration [20,21,22].

Other than platelets releasing granule contents, which directly stimulate hepatocyte proliferation [23,24] or the transfer of RNA by platelets, which promotes hepatocyte proliferation either by the translation of mRNA or by the action of regulatory RNAs, a potential mechanism underlying platelet-mediated liver regeneration could involve facilitation of the inflammatory response [25]. In this instance, platelets would have the functionality of attracting inflammatory cells [26,27,28,29,30,31]. Recent accumulating evidence points to platelets playing a major role in stem and progenitor cell homing along the vascular interface in the scenario of regeneration [29,32,33]. Furthermore, synergistic effects of platelets and progenitor cells have been shown for regeneration following ischemic vascular disease [34]. Recently, platelets have been reported to play a role in the repair following hepatic damage by activating the hematopoietic-vascular niche to generate pro-regenerative endothelial paracrine/angiocrine factors [35]. We have previously shown in a rodent model that BMSC mobilization by CD39 facilitates liver regeneration and proliferation after partial hepatectomy [1].

Here, we propose that platelets play a role in modulating BMSC interactions with the regenerating hepatic vasculature. Platelets were tested for their potential to boost CD133^+^BMSC recruitment to LSECs. The underlying receptor–ligand interactions of platelet/endothelial cell interrelations were characterized under flow conditions. Furthermore, we evaluated the impact of platelet-CD133^+^BMSC-interactions on homing of these stem cells to the isolated single pass perfused rat liver (IPRL) following warm ischemia.

## 2. Results

### 2.1. Platelets Augment CD133^+^BMSC Adhesion to Human Micro-Endothelium under Shear Stress In Vitro

Since it was previously shown that platelets have a significant impact for hepatic injury and regeneration, especially on LSEC [10,11,12,13,14,20,21], we further investigated CD133^+^BMSC-endothelium interactions with respect to the role of platelets for local vascular homing under flow conditions. We established a live cell imaging system (BIOFLUX) [36], in which human micro vasculature endothelial cells (HMEC-1) were grown in capillaries of the BIOFLUX system under flow conditions and subsequently co-cultivated with human platelet rich plasma (hPRP) and primary human CD133^+^BMSC under different conditions before infusion to the BIOFLUX system. Under shear stress levels of 1.0 dyne, we could demonstrate that hPRP-co-culture increases the number of adhering CD133^+^BMSC to human micro-endothelium significantly (*p* < 0.01) by a mean of 2.6-fold (+/−1.5) if contrasted to hPPP (Figure 1a).

### 2.2. The Relevance of the P-Selectin/PSGL-1-Axis for the Effect of Platelets to Improve CD133^+^BMSC Adhesion to Human Micro-Endothelium

To investigate the role specific receptor-ligand interactions for the effect of platelets on the capacity of human CD133^+^BMSC to adhere along human EC under flow, we first examined P-selectin and its ligand PSGL-1 to that respect. Statistically as a trend (*p* = 0.067) pre-incubation of hPRP with the P-selectin-specific antagonist KF38789 reduced adhesion levels when contrasted to non-antagonised hPRP-co-culture of CD133^+^BMSC and to a similar level observed for platelet poor conditions (48.3 +/− 24.4% vs. 39.3 +/− 26.1%; Figure 1b) in all paired experiments performed in this study. Likewise, PSGL-1-blockage on CD133^+^BMSC revealed a reducing effect on the platelet depending augmentation of adhesion of CD133^+^BMSC to EC under shear stress (*p* < 0.01; Figure 1c). Next, we evaluated the effect of PECAM-1 on EC to bind platelets. Inhibition of PECAM, by either pre-incubation of EC with PECAM-1-blocking antibody (Figure 1d) or with co-infused recombinant soluble PECAM-1 (Figure 1e) had no modulating effect on platelet promoted CD133^+^BMSC adhesion to EC. As we demonstrated the SDF-1/CXCR4 interaction to be relevant for systemic mobilisation of CD133^+^BMSC in the course of clinical liver regeneration subsequent to parenchymal loss [6], we tested the CXCR4-inhibitor (AMD3100) for a modulatory impact on platelet promoted adhesion of CD133^+^BMSC to HMEC1. However, there was no modulation of the adhesion rate of CD133^+^BMSC to HMEC-1 subsequent to co-incubation with AMD3100 (Figure 1f). These results indicate that PSGL-1 on BMSC interacting with its receptor P-selectin on platelets might be important for the augmentation of platelet-mediated CD133^+^BMSC-homing along EC. In contrast, PECAM-1 and the SDF-1/CXCR4-axis seemed to play only a minor part in that scenario.

### 2.3. Platelet Promoting Effect In Vitro on CD133^+^BMSC Adhesion to Endothelium is Conserved for Rodent Micro Endothelium and LSEC Independent of Further Stimulation

Next, we tested the impact of platelets in an allogeneic rodent equivalent of our human shear-stress co-culture model. Murine platelets (mPRP) had a similar adhesive enhancing effect for mouse (m) CD133^+^BMSC to murine dermal micro-endothelial cells (dMEC) when contrasted to platelet-poor conditions (mPRP vs. mPPP 1.44-fold (+/− 0.17); *p* < 0.01, Figure 2a). Further, stimulation of platelets with the strong platelet activator ADP exhibited a little more pronounced effect on CD133^+^BMSC adhesion (*p* < 0.001; Figure 2b). However, when directly compared to non-activated platelet co-incubation, we noted only a non-significant trend (+ ADP vs. − ADP; *p* = 0.072). To prove the platelet effect in the same setting for hepatic sinusoidal endothelial cells, we utilized mouse liver sinusoidal endothelial cell (mLSEC) in our flow chamber system and observed a comparable positive platelet effect of mCD133^+^BMSC adhesion to mLSEC (1.31 fold (+/− 0.09); *p* < 0.05; Figure 2c).

### 2.4. Platelets Promote Hepatic Homing and Extravasation of CD133^+^BMSC in the Course of Reperfusion Subsequent to Warm Ischemia

Next, we questioned the effect of platelets on the hepatic homing capacity of CD133^+^BMSC to the micro-architecture of the rodent liver following warm ischemia tested in a single pass xenogeneic IPRL system (Figure 3). CD133^+^BMSC were tracked by in situ video microscopy to detect intra-sinusoidal adhesion to LSEC as well as co-localization of BMSC to the hepatic parenchyma as a marker of extravasation (sample video-microscopy image: Figure 4a). 

In order to prove and quantify the effect of platelets for BMSC-homing following warm liver ischemia and reperfusion, CD133^+^BMSC injection to the IPRL was performed in three experimental groups: Group I: CD133^+^BMSC only (-PRP), Group II: CD133^+^BMSC after platelet pre-infusion of the rat liver with platelets (post PRP) and Group III: CD133^+^BMSC after co-incubation with platelets (with PRP). After warm liver ischemia the total number of homing CD133^+^BMSC demonstrated a non-significant trend towards higher numbers of homing BMSC in groups II and III if contrasted to group I (each *n* = 5; Figure 4b). Compared to group I, the level of CD133^+^BMSC-extravasation, represented as relation of parenchymal- to intra-sinusoidal-located BMSCs, was significantly increased in group II and III (*p* < 0.05; Figure 4c). The positive trend of co-incubation with CD133^+^BMSC if contrasted to pre-infused platelets prior to stem-cell-infusion was statistically not significant. Subsequently, absolute numbers of extra sinusoidally detectable CD133^+^BMSC were significantly increased in group II (9.9 +/− 2.6 cells) and group III (13.3 +/− 5.7 cells) per 10 high power fields when contrasted to group I (4.2 +/− 2.6; *p* < 0.05; Figure 4d).

### 2.5. Localization of Human CD133^+^BMSC Homing during Hepatic Warm Ischemia Reperfusion Injury

Following warm ischemia and subsequent infusion of human CD133^+^BMSC via the portal vein, immunofluorescence staining and confocal microscopy analysis were applied to characterize the homing of CD133^+^BMSC to the rat liver. Serial liver tissue slides taken from the rat liver subsequent to 2.5 h of continued reperfusion were stained for different cell-specific marker proteins as well as for human CD45, which was specific for the infused CD133^+^BMSC. CD133^+^BMSC with a normal nucleus-to-cytoplasm ratio were found within the liver tissue after 2.5 h of perfusion (Figure 5a). 

CD133^+^BMSC were not only located in areas of connective tissue, but were especially found in close contact to hepatocytes, evidenced by their proximity to F-actin- and Ntcp-positive hepatocytes (Figure 5b–d). BMSCs entered the parenchyma subsequent to extravasation located with distance to hepatic endothelial and sinusoidal structures, detected by co-staining for the rat endothelial cell marker RECA-1 (Figure 5e,f) However, a certain disruption of the endothelial layer on the level of sinusoids to some extend due to warm ischemic hepatic damage cannot be excluded. CD133^+^BMSC were detected on the acinar and anatomical lobular level in hepatic zones 1 and 2, whereas cells were rarely detectable in the peri-central-venular zone 3, indicated by zone 3 glutamine synthetases co-staining (Figure 5G,H), suggesting that the cells exited the blood along most parts of the sinusoidal conduit.

## 3. Discussion

In the present study, we provide evidence that platelets promote CD133^+^BMSC interaction with human endothelium under flow conditions dependent on P-selectin/PSGL-1 interactions. In the autologous rodent model, we confirmed the adhesion promoting effect of platelets under shear stress to micro-endothelium as well as to LSEC in vitro. Furthermore, in a chimeric human to rat model, co-administration of platelets with CD133^+^BMSC following hepatic warm ischemia increased local vascular homing and extravasation of the stem cells within the liver.

The contribution of BMSCs to liver regeneration is complex as some authors hypothesize that stem cells reconstitute the regenerating liver by transdifferentiation into primary hepatocytes [37,38,39,40,41,42,43]. Contrarily, this has been challenged in subsequent reports [44]. Others advocate that extrahepatic BMSC undergo cell fusion [45,46] or function as external or anti-inflammatory regulators required for successful liver restoration [38,47,48]. Overall, the contribution of hematopoietic stem cells (HSC) to liver repair seems generally to be related to the presence and severity of liver injury. BMSC have the potential to enter different organs from the circulation to induce organ repair. However, the exact mechanisms and cellular processes remain unclear. It has been observed that extrahepatic CD133^+^HSC mobilization is augmented in response to large liver resections that induce adequate liver regeneration experimentally [1] and clinically [3,6,49].

We have previously provided evidence for cytokines and chemoattractants to serve as mediators to enable adequate CD133^+^BMSC mobilization. This promotes regeneration after large liver resections and levels of mobilization that correlate with the magnitude of clinical liver regeneration after hepatectomy [6]. In the same study, evidence was provided for liver injury to induce the expression of signaling mediators that facilitate the recruitment and homing of HSC to the damaged liver [6,50,51,52,53,54,55,56,57,58].

The observed platelet mediated augmentation of CD133^+^BMSC adhesion to the sinusoidal micro-vasculature in this study may be facilitated by cell-cell-interactions. Latter was demonstrated for human peripheral blood derived CD133^+^ endothelial progenitor cells (EPC), monocytes and CD133^+^BMSC respectively when co-incubated with platelets [29,30]. We noted this adhesion enhancing effect under shear stress conditions closer to the physiological environment of the micro-vasculature of the liver. Our data are in line with reports on platelet interaction with neutrophils, lymphocytes, CD39^+^ cells and CD133^+^EPC respectively under flow [27,32,59].

The functional blockage of P-selectin on platelets mainly abolished the enhancing effect of platelets binding to human micro-EC. This observation goes along with reports on P-selectin to play a critical role for binding leucocytes along vasculature like monocytes, CD34^+^ cells, bone marrow mesenchymal stem cells and EPC [29,32,59]. The impact of P-selectin for hepatic platelet and leucocytes homing was shown in an in vivo warm liver ischemia model, demonstrating polymorphonuclear leucocytes adhesion to be significantly decreased in P-selectin-deficient mice if contrasted to wild-type animals [28]. PSGL-1 antigen is known to be expressed on hematopoietic stem cells and other lines of leukocytes as the major ligand for P-selectin [60,61]. Here, we demonstrated that interactions with PSGL-1 on CD133^+^BMSC is critical for the platelet-mediated boost of adhesion along EC as it was observed for EPC binding to vascular injury sites [59]. PECAM-1 is known to be expressed by a wide range of cell lines and types including leucocytes, monocytes, hematopoietic stem cells, platelets, and endothelial cells [62,63,64,65]. This adhesion molecule was demonstrated to play a role for recruitment, migration on and transmigration along vasculature for leucocytes like neutrophils and monocytes [66,67]. We observed no substantial role for PECAM-1 for a modulatory effect on CD133^+^BMSC homing to vasculature under flow. Still, we cannot exclude an effect of PECAM-1 for endothelial transmigration processes as described for various types of vasculature [68]. Among other factors located on or known to be derived from platelets that are discussed to play a role for hematopoietic stem cell and progenitor cell homing to locally bound platelets at sites of vascular and heart injury is stroma derived factor 1 (SDF-1) [69,70,71]. The lack of a modulatory effect of CXCR4-inhibition for vascular homing of CD133^+^BMSC under shear force here suggests the SDF-1/CXCR4-axis plays a minor role in local endothelial homing in contrast to our previous report on SDF-1 to be relevant for peripheral CD133^+^BMSC-mobilisation in the scenario of pronounced hepatic regeneration [6].

The ameliorating effect of human platelets for human CD133^+^BMSC under flow in a rodent model with murine micro EC and LSEC is in agreement with the observations of Lalor et al., who demonstrated in a study on hepatic vasculature deploying static as well as flow conditions that platelets bind to LSEC even more effectively than to human umbilical vein endothelial cells (HUVEC). Under shear stress, platelets promoted leucocyte adhesion to LSEC which was shown to be in part P-selectin-dependent [27]. In the rodent setting, pre-stimulation of platelets prior to co-incubation with CD133^+^BMSC resulted in a non-relevant effect on platelet-dependent improvement of CD133^+^BMSC homing to EC. A former study demonstrated significantly enhanced interaction of platelets with endothelial progenitor cells after pre-stimulation with various platelet activators such as ADP [59]. A study on whole blood with thrombin receptor activating peptide as platelet stimulator revealed superior platelet activation upon broad leucocyte platelet interaction [72]. Other platelet stimulators were not tested to that respect. Our group previously demonstrated a regulatory effect of CD133^+^BMSC for ADP-dependent platelets aggregation in co-incubation of these two cell-types characterized as NTPDase1 (CD39) dependent [8]. This is in line with a report on eosinophils to directly inhibit platelet aggregation among others due to ADP-stimulation [73]. This mechanism of interaction of stem cells and platelets in the course of regenerative interactions with vasculature paralleled by a control of excessive thrombogenic processes may add to explain the missing effect of platelet stimulation in the here presented study.

Human CD133^+^BMSC homing and co-localization to rat hepatocytes in the micro-architecture of the reperfused liver subsequent to warm ischemia goes along with in vivo observations in rodent models of acute hepatic injury. Here, bone marrow derived mesenchymal and hematopoietic stem cells locate to the damaged liver promoting regeneration processes [74,75]. Further verification of the salutary effect of platelets was facilitated employing in situ video microscopy as previously utilized for other rodent scenarios to track sinusoidal leucocyte homing and extravasation subsequent to warm hepatic ischemia [76,77]. To facilitate tracing of infused cells we utilized GFP-tagging of the CD133^+^BMSC. Such fluorescence staining concepts are routinely utilized in models of hepatic damage and subsequent bone marrow stem cell treatment in the rat for tracking applicated cells [78,79,80]. Although tested under plasma-free conditions, the selected xeno-model of human CD133^+^BMSC transfused to rat livers may bear some weakness due to xenogeneic effects. However, our observation of pronounced extravasation from the sinusoidal space towards parenchymal structures secondary to platelet interference is in line with previous studies on platelet interaction with neutrophils and monocytes respectively at sites of inflammation that demonstrated accelerated leucocyte adhesion followed by eased trans-endothelial migration due to P-selectin/PSGL-1 interaction dependent platelet intervention [81,82]. Moreover, in a murine skin model, platelets exhibited a significant role for trafficking of mesenchymal stem cells to the extravascular space at the site of inflammation for both, venules as well as microendothelial capillaries [83]. In the therapeutic approach, platelet-stem-cell linking antibodies have been reported to improve homing of administrated stem cells leading in amelioration of their regenerating effect in a model of cardiac ischemia/reperfusion injury [84].

We have previously successfully applied human autologous CD133^+^HSC in a clinical scenario by intraportal administration following portal venous embolization of right liver segments to expand left lateral hepatic segments prior to extended liver resection [2,3,4]. Although the exact mechanisms by which extrahepatic stem cells and their progenitor cells promote liver regeneration are not fully elucidated [45,85,86], the liver proliferative effect of therapeutic stem cell applications in the clinical and pre-clinical scenarios of ischemia or hepatic volume loss may in part be due to local accumulation of administered BMSC prior to physiologically chemottractant-driven mobilization and homing. Our observations here provide a key mechanistic distinction to deepen our understanding of the proposed model of hepatic homing of CD133^+^BMSC subsequent to liver injury with platelets promoting sinusoidal adhesion and subsequent transmigration to the extra-sinusoidal space, both forwarded by P-selectin/PSGL-1 interaction (Figure 6). Alternatively, platelets may bind to the matrix of the space of Disse in areas of disturbed sinusoidal vascular disintegrity with subsequent promotion of CD133^+^BMSC accumulating to the space of Disse. Such platelet-space of Disse interactions were demonstrated for ischemia-reperfusion scenarios as observed subsequent to liver transplantation before (see [87,88]). Further studies on the diverse variants of hepatic damage need to be conducted in order to elucidate the exact mechanisms of platelets to support extravasation of homing CD133^+^BMSC to the liver.

In light of the data presented here, this study may lead to new approaches to accelerate efficacy of BMSC treatment strategies in the regeneration, protection, and treatment of various liver diseases. Platelets might be a useful strategy to that respect in order to maximize the effectiveness of therapeutically applied BMSC. Various constellations of acute and chronic hepatic injury may benefit from such treatment concepts based on BMSC. These may include liver transplantation, conditions after extensive hepatic resection, hepatitis, and acute-on-chronic episodes in the context of liver cirrhosis. However, further studies are needed to explore the exact mechanisms of platelet supported local homing of BMSC into the liver and resultant regeneration and proliferation.

## 4. Materials and Methods

### 4.1. Cells and Cell Isolation

Primary human CD133^+^BMSC were purified from bone marrow aspirates from patients undergoing abdominal surgery following written informed consent with approval from the local ethics committee (Ethics Committee of the Heinrich-Heine-University, Duesseldorf, Germany; approval no. 2852 and no. 2853, 2 February 2007). All methods were carried out in accordance with relevant guidelines and regulations. For purification, magnetic activated cell sorting (Miltenyi Biotec, Bergisch Gladbach, Germany) according to manufacturer’s instructions was utilized. Purity was assessed by FACS analysis of each preparation using a BD FACSCanto (Becton Dickinson, Heidelberg, Germany) flow cytometry system. The following antibodies were used: human CD133/2-PE (293C3, Miltenyi Biotec, Bergisch Gladbach, Germany), human CD45-APC (APC anti-human CD45, Becton Dickinson, Heidelberg, Germany), and human CD 34-PE-Cy7 (PE-Cy7 mouse anti-human CD34, Becton Dickinson, Heidelberg, Germany). Primary murine CD133^+^BMSC were purified from bone marrow flushed from murine tibiae and femori (male C57BL/6 mice) utilizing phase separation for mononuclear cells and adjacent FACSorting (anti-Prominin-1-PE, Miltenyi Biotec, Bergisch Gladbach, Germany; anti-mouse CD34-eFluor 660, eBioscience, Dreieich, Germany; anti-CD 45-FITC, Miltenyi Biotec, Bergisch Gladbach, Germany). The animal research was performed conform to national guidelines and with approval of the local committee. All experimental protocols were approved by the named licensing committee of the District Government Düsseldorf, Germany (internal no O/56/06 ). 

For co-cultivation experiments with thrombocytes, platelet rich plasma (PRP) and platelet poor plasma (PPP) was prepared from 1012 mL of blood from respective BMSC donors by blood centrifugation. For PRP, the blood was centrifuged for 10 min at 20° at 180× *g* before PRP was collected from the supernatant. For PPP, the blood was centrifuged for 5 min at 20° at 600× *g* before PPP was collected from the supernatant. Human dermal microvascular endothelial cells (HMEC-1) were kindly provided by the Department of Anesthesiology, Intensive Care and Pain Medicine, Experimental and Clinical Hemostasis, University of Muenster, Germany and cultivated in Endothelial Medium (PAA, # U15-002) with Glutamine, Penicillin/Streptomycin and growth factors Hydrocortison (1 µg/mL) and endothelial growth factor (10 ng/mL) at 37 °C and 5% CO_2_. Mouse primary dermal microvasculature endothelial cells (dMEC) and mouse hepatic sinusoidal endothelial cells (mLSEC) were commercially available (C57BL/6, PeloBiotech, Planegg, Germany) and cultivated in Endothelial Cell Medium (PeloBiotech, Planegg, Germany; # PB-M-1168) at 37 °C and 5% CO_2_.

### 4.2. Co-Culture Live Cell Assay

Human and mouse endothelial cells (HMEC-1, dMEC, and mLSEC) were cultured in fibronectine-coated capillaries of a live cell imaging system (BIOFLUX 200, Fluxion). We established a heterologous live cell imaging system for co-cultivation of human and mouse endothelial cells respectively, co-cultured under shear stress with CD133^+^BMSC and platelet rich or poor plasma as control prepared from respective BMSC donors under different test conditions before infusion to the BIOFLUX system. We tested in both species to evaluate conservation of effects across species to justify coming preclinical studies and ones in systems modified for relevant genes. The BIOFLUX System contains microfluidic flow channels that connect two wells of a 14-well microtiter plate. Air pressure is applied to induce flow conditions in capillaries at a physiological shear force of 1 dyn/cm^2^. All experiments were performed in duplicate testing treatment versus control (DMSO/H_2_O) conditions parallel with the same preparation of BMSC for 1 h at 1 dyn/cm^2^ followed by quantification of adherent CD133^+^BMSC. In human experiments, we performed different blocking experiments by pre-incubation of different cell types with inhibitors or blocking antibodies: 20 min pre-incubation of platelets with the selective P-selectin antagonist small-molecule KF38789 (Tocris, Bio-Techne GmbH, Wiesbaden, Germany; 10 µM) [89] at room temperature, 10 min pre-incubation of CD133^+^BMSC with 10 µg/mL PSGL-1 antagonist for SDF-1 interaction IM2090 (Beckman Coulter, Krefeld, Germany; Clone 3E2-25-5-PL1) at 37 °C, 20 min pre-incubation of HMEC-1 with 10 µg/mL PECAM-1 blocking antibody mPECAM-1.3 IgG (kindly provided by Prof. P. Newman, Blood Research Institute, Milwaukee, WI, USA) at 37 °C. As a second PECAM-1 inhibiting strategy, we performed co-culture experiments under shear force co-incubating with recombinant truncated PECAM-1 protein (R&D Systems, Bio-Techne GmbH, Wiesbaden, Germany; 3.0 μg/mL). This soluble form encompasses the extracellular fraction of PECAM-1 being in competition with cell bound PECAM-1 for binding. To evaluate the role of the SDF-1/CXCR4 axis we performed experiments under CXCR4 inhibiting conditions, pre-incubating CD133^+^BMSC with the CXCR4-inhibitor AMD3100 (5 µg/mL) as previously performed [6]. To test the effect of pre-stimulation of platelets for adhesion of CD133^+^mBMSC, murine PRP was pre-stimulated with 0.5 µM ADP monitored by aggregometry utilizing the Born-light-transmission aggregometry (LTA) method in a Lumi-Aggregometer (Chronolog). Latter prevented over-stimulation with clot-formation.

### 4.3. Isolated Perfused Rat Liver (IPRL)

We established a single pass, xenogeneic in situ IPRL model-system in combination with video-assisted in situ imaging to evaluate an impact of platelets for homing of CD133^+^BMSC to the liver following warm ischemia ex vivo (Figure 3). A serum-free single pass xeno-perfusion model of human platelets and BMSC, respectively transfused to the rat liver was selected to test human blood-born components with the liver micro architecture, as a test system incorporating human liver is not available at present. In this system, human primary CD133^+^BMSC, human platelet rich plasma (PRP) and platelet poor plasma (PPP) were prepared from the same donors. Rat livers of male Wistar rats (120–160 g) were linked to an isolated perfused rat liver model (IPRL). We performed portal vein cannulation of the rat liver left in situ and secured hepatic venous flow via cannulation of the vena cava superior. Additional cannulation of the ductus choledochus allowed drainage of bile. After stable flow was established with Krebs–Henseleit Buffer at 37 °C at a flow rate of 250 µL/min a warm hepatic ischemia was achieved by pausing the hepatic flow for 30 min. The ischemic phase was followed by a reperfusion period of 3 h length. Hereafter, 2 × 10E5 human CD133^+^BMSC were infused as a bolus to the liver perfusing portal venous flow. 

To analyze the impact of platelets for homing of CD133^+^BMSC to the rat liver we determined three experimental groups and performed five independent IPRL experiments in each group. The following groups were analyzed. In Group I, 2 × 10E5 CD133^+^BMSC were resuspended in Krebs-Henseleit–Buffer and directly injected to the IPRL (CD133^+^BMSC -PRP). In group II, primary injection of 3 mL PRP to the rat liver and subsequent application of 2 × 10E5 CD133BMSC resuspended in Krebs–Henseleit Buffer to the IPRL (CD133^+^BMSC following PRP). In Group III, 15 min incubation of 2 × 10E5 CD133^+^BMSC in 1 mL PRP and subsequent injection to the IPRL via 1 mL syringe (CD133^+^BMSC co-incubated with PRP). Quantification of total hepatic homing and the proportion of extravasation of BMSC were realized by fluorescence cell membrane labeling (PKH67 Green, Sigma Aldrich, Munich, Germany) and in situ imaging of the IPRL (Figure 2). Extra-sinusoidal (parenchymatous) vs. sinusoidal (intra-vascular) CD133^+^BMSC were analyzed after liver passage. Fluorescence staining of CD133^+^BMSC with PKH67 Green was performed to analyze amount and localization of human BMSC homing to the rat liver by fluorescence microscopy of 10 different visual fields. Quantitative image analyses were performed blinded to the investigator concerning treatment group. 

### 4.4. Confocal Fluorescence Microscopy and In Situ Imaging

Characterization of human CD133^+^BMSC homing to the rat liver was carried out by immunofluorescence staining and (confocal) microscopy analysis of serial liver tissue slides. Immunofluorescence staining and confocal fluorescence microscopy (20× magnification) was performed for human CD45 (human CD45: mouse monoclonal IgG, clone 35-Z6, Santa Cruz, Heidelberg, Germany) conjugated to Alexa Fluor 488 goat anti-mouse (Biolegend, London, United Kingdom), for the endothelial cell marker RECA-1 [90], for the hepatocytes-specific sodium taurocholate cotransporting polypeptide (Ntcp) with a rabbit anti-rat Ntcp (K4) [91], rat Phalloidin (Phalloidin-TRITC) staining of F-Aktin filaments in the rat liver and additional nuclear DAPI-staining. Extravasation of CD133^+^BMSC was characterized by human-specific CD45 immunofluorescence staining. To differentiate hepatic zone distribution of zone 1 and 2 vs. 3, we detected for tyrosine synthetase, known to predominantely being detectable peri-central-venular in zone 3 hepatocytes of rat livers [92]. Cells located to the parenchymal bars delineating sinusoids were determined as “parenchymal”. Cells located to the free sinusoids were determined as sinusoidal. Extravasation was similarly demonstrated in our study as performed by others for tumor cells utilizing in-situ-imaging in the rat-liver [93].

### 4.5. Statistics

Statistical analysis and graphing were performed using MS Excel, Systat 13 and SigmaPlot 14. All results are expressed as mean ± standard deviation. Statistical significance was determined by Student’s two sided and paired t test respectively and significance was defined as * *p* < 0.05, ** *p* < 0.01, *** *p* < 0.001.

## Figures and Tables

**Figure 1 ijms-21-06431-f001:**
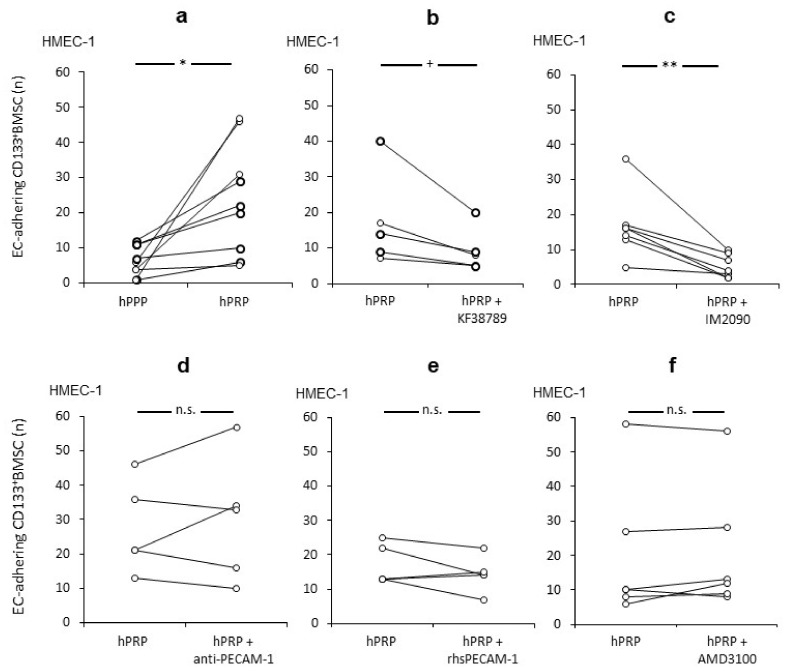
P-selectin/PSGL-1 dependent platelet interactions with CD133^+^BMSC promote adhesion to human micro-EC under shear stress. Adherence of CD133^+^BMSC to human micro endothelial cells (HMEC-1) co-incubated with human platelet rich plasma (hPRP) was tested by pairs under different conditions: control and treatment at a time. (**a**) Increased CD133^+^BMSC adherence with hPRP when compared to platelet poor plasma (hPPP). (**b**,**c**) Both Pre-incubation of platelets with P-selectin-inhibitor KF38789 and CD133^+^BMSC with PSGL-1 antagonist IM2090 revealed a reduction of adherence of CD133^+^ BMSC. (**d**–**f**): Co-incubation with PECAM-1-blocking antibody mPECAM-1.3 IgG (anti-PECAM-1), recombinant soluble human PECAM-1 (rhsPECAM-1) and CXCR4-inhibitor for SDF-1 interaction AMD3100 respectively lacked a modulating effect on CD133^+^BMSC for adherence to HMEC-1. Paired *t*-test: * *p* < 0.05; ** *p* < 0.01; ^+^
*p* = 0.067; n.s. *p* > 0.1.

**Figure 2 ijms-21-06431-f002:**
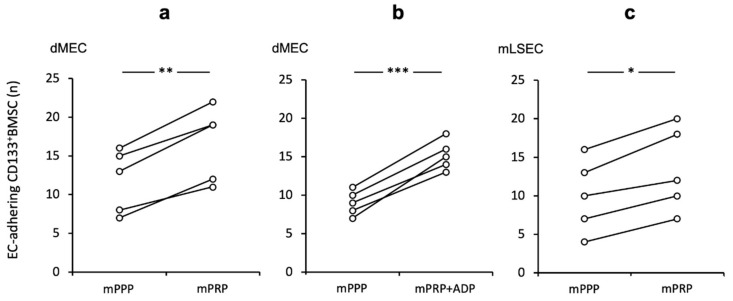
Augmented CD133^+^BMSC adherence to endothelium subsequent to platelet co-incubation in a murine shear-stress model Adherence of CD133^+^BMSC to murine endothelial cells co-incubated with mouse platelet rich plasma (mPRP) was tested by pairs under different conditions: control and treatment at a time. (**a**) Significant increase in adhering mouse CD133^+^BMSC to mouse dermal micro-endothelial cells (dMEC) with mPRP when contrasted to mPPP (mouse platelet poor plasma). (**b**) Platelet activation by ADP did not further enhance the mPRP effect. (**c**) Augmenting platelet effect on mCD133^+^BMSC adhesion to mLSEC. Experiments were performed at shear levels of 1.0 dyne/cm^2^ followed by quantification of adherent CD133^+^BMSC in % of mPPP control conditions. Paired *t*-test: * *p* < 0.05; ** *p* < 0.01; *** *p* < 0.001.

**Figure 3 ijms-21-06431-f003:**
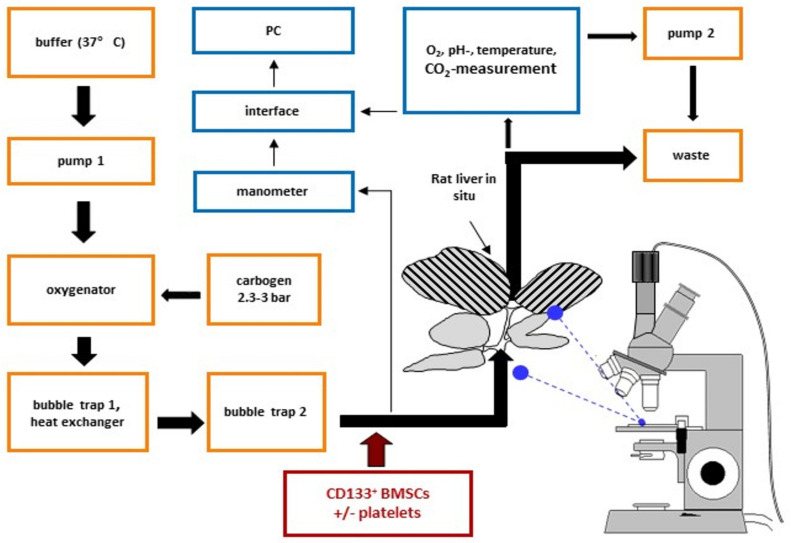
Ex vivo xenogeneic IPRL model with video-assisted in situ imaging. Direction of arrows indicates buffer and cell flow in the in situ system. The black shades stripes within the analyzed liver represent 70% hepatectomy. Portal vein of the rat liver was cannulated and left in situ and liver was drained via the cannulation of the vena cava superior.

**Figure 4 ijms-21-06431-f004:**
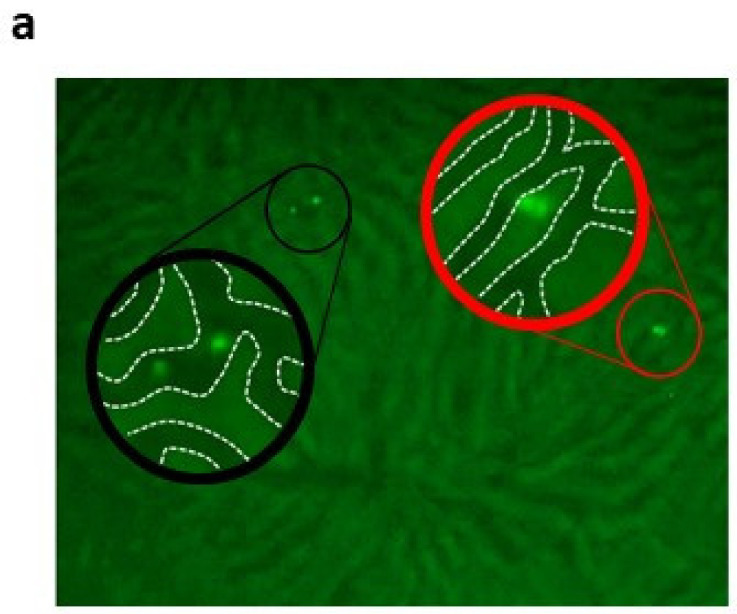
Platelets increase CD133^+^BMSC binding and extravasation in an ischemia and reperfusion injury model. CD133^+^BMSC were infused to the rat liver after 180 min of reperfusion. In Situ Imaging by fluorescence microscopy of 10 different visual fields. (**a**) In situ Imaging: Extra-sinusoidal (parenchymatous, red circle) vs. sinusoidal (intra-vascular, black circle) CD133^+^BMSC after liver passage. Membranes of CD133^+^BMSC were labeled prior to in situ perfusion with PKH67 Green to analyze amount and localization of human BMSC homing to the rat liver by fluorescence microscopy of at least 10 different visual fields. (**b**) IPRL: Increase of absolute numbers of CD133^+^BMSC homing to the liver (CD133^+^BMSC cell count per 10 visual fields) without platelet co-treatment (column 1; -PRP) versus CD133^+^BMSC pre-infusion of PRP and subsequent infusion of stem cells to the rat liver (column 2; postPRP) versus CD133^+^BMSC co-incubated with PRP and subsequent co-infusion (column 3; withPRP). (**c**) Relation of parenchymal versus vascular located CD133^+^BMSC in these 3 groups. (**d**) Absolute numbers of CD133^+^BMSC located to the extra-sinusoidal parenchyma. * *p* < 0.05; n.s., not significant.

**Figure 5 ijms-21-06431-f005:**
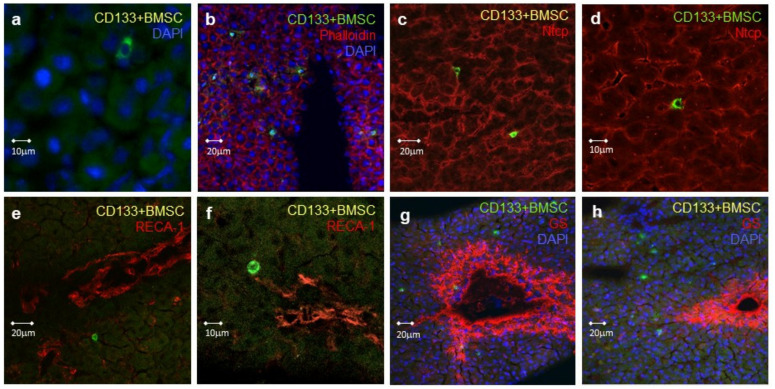
Characterization of hepatic homing of CD133^+^BMSC in IPRL by confocal fluorescence microscopy. Rat livers were perfused with CD133^+^BMSC, which were stained by anti-CD45 antibodies (green). (**a**) Cell nuclei were stained by DAPI (blue). (**b**) Demonstration of CD133^+^BMSC (green) co-localization with hepatocytes visualized by actin-staining of hepatocytes with TRITC-labelled phalloidin (red). Nuclear staining with DAPI (blue). (**c**,**d**) CD133^+^BMSC (green) co-localization to hepatocytes, specifically detected by Sodium taurochlate cotransporting protein- (Ntcp-) staining (red). Ntcp is exclusively expressed in hepatocytes. (**e**,**f**) Confirmation of CD133^+^BMSC (green) extravasation with distance to LSEC. Latter is specifically detected by the endothelial cell marker RECA-1 (red). (**g**,**h**) CD133^+^BMSC (green) are predominantly localized to hepatic zone 1 and 2 but rarely co-located to the inner zone 3 indicated by staining of glutamine synthetase (GS; red). The latter is mainly expressed in hepatocytes of zone 3 close to the central venule of the acinus. Nuclear staining with DAPI (blue).

**Figure 6 ijms-21-06431-f006:**
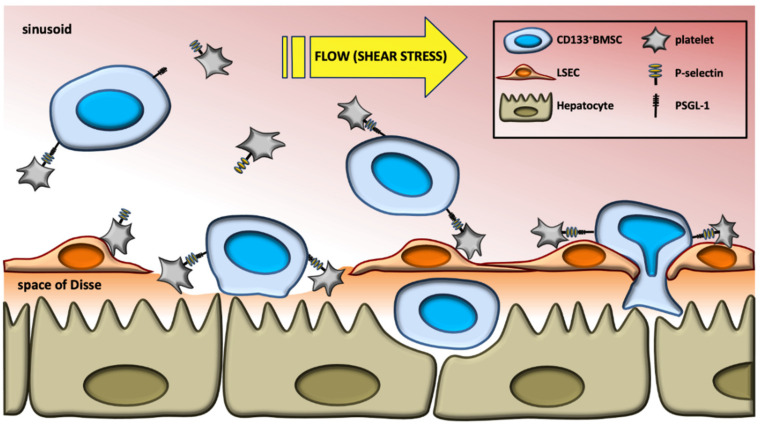
Model of platelets to increase hepatic homing of CD133^+^BMSC to the injured liver. In response to shear stress in the liver, platelets promote sinusoidal adhesion of CD133^+^BMSC and subsequent transmigration to the extra-sinusoidal space, both forwarded by P-selectin/PSGL-1 interaction.

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
