# Peer review of "Platelets Boost Recruitment of CD133^+^ Bone Marrow Stem Cells to Endothelium and the Rodent Liver—The Role of P-Selectin/PSGL-1 Interactions"

_ijms, 2020, doi:10.3390/ijms21176431_

Round 1

Reviewer 1 Report

The article “Platelets boost recruitment of CD133+ bone marrow stem cells to endothelium and the rodent liver – the role of P-selectin/PSGL-1 interactions” is an interesting research, but I have the following questions.

1) Please describe the way how PPP and PRP solution creates in the materials and methods. How much platelet is included in PPP or PRP? Is there any other difference except platelet count?

2) What is the dMEC “d” in Figure 2? Is it different from mMEC? Did the authors use human-derived ones, as written in Figure2? Or, did the authors use mouse-derived “mPPP or mPRP” rather than “hPPP or hPRP”?

3) I wonder reagents such as KF38789 and IM2090 cannot be used in mouse-derived endothelial cells. If they can be used, could the authors do the same experiments in Figure 2 as that in Figure 1?

4) What kind of clinical background did the authors assume in Group II and Group III of a chimeric human to rat model? How do the differences between Group II and III results support the hypothesis in Figure 6?

5) I think that most of rat SECs will be lost by 30 minutes of warm ischemia and reperfusion. In fact, most the stain around the sinusoid seems to be lost in the RECA-1 staining in Figure 5. If SECs were missing, wouldn't it be difficult for the results to support the hypothesis in Figure 6?

Author Response

Please see in the attachment

Reviewer 2 Report

The work is interesting and well designed. Some suggestions, you can add the conclusions in a section. and perspectives of clinical utility. Thank you

They can be specific in their model, in relation to which they mean that it can protect and treat various liver diseases.

Your model and results, as you can expect or compare with models in animals (partial hepatectomy) or in humans (liver transplantation)

Author Response

We thank the reviewer for these positive comments. According to those suggestions, we have modified the conclusions in the discussion part (last paragraph). This now reads as following: “In light of the data presented here, this study may lead to new approaches to accelerate efficacy of BMSC treatment strategies in regeneration, protection and treatment of various liver diseases. Platelets might be a useful strategy to that respect in order to maximize the effectiveness of therapeutically applied BMSC. Various constellations of acute and chronic hepatic injury may benefit from such treatment concepts based on BMSC. These may include liver transplantation, conditions after extensive hepatic resection, hepatitis and acute-on-chronic episodes in the context of liver cirrhosis. However, further studies are needed to explore the exact mechanisms of platelet supported local homing of BMSC into the liver and resultant regeneration and proliferation.”

Round 2

Reviewer 1 Report

The authors answered all the questions.